# Reduced Cerebellar Volume in Term Infants with Complex Congenital Heart Disease: Correlation with Postnatal Growth Measurements

**DOI:** 10.3390/diagnostics12071644

**Published:** 2022-07-06

**Authors:** Rafael Ceschin, Alexandria Zahner, William Reynolds, Nancy Beluk, Ashok Panigrahy

**Affiliations:** 1Department of Radiology, University of Pittsburgh School of Medicine, Pittsburgh, PA 15213, USA; panigrahya@upmc.edu; 2Department of Biomedical Informatics, University of Pittsburgh School of Medicine, Pittsburgh, PA 15213, USA; william.reynolds@chp.edu; 3Department of Radiology, UPMC Children’s Hospital of Pittsburgh, Pittsburgh, PA 15224, USA; alexandria.zahner@chp.edu (A.Z.); beluknh@upmc.edu (N.B.)

**Keywords:** congenital heart disease, Neonatal Imaging, structural MRI, postnatal growth

## Abstract

Aberrant cerebellar development and the associated neurocognitive deficits has been postulated in infants with congenital heart disease (CHD). Our objective is to investigate the effect of postnatal head and somatic growth on cerebellar development in neonates with CHD. We compared term-born neonates with a history of CHD with a cohort of preterm-born neonates, two cohorts at similar risk for neurodevelopment impairment, in order to determine if they are similarly affected in the early developmental period. **Study Design:** 51 preterms-born healthy neonates, 62 term-born CHD neonates, and 54 term-born healthy neonates underwent a brain MRI with volumetric imaging. Cerebellar volumes were extracted through an automated segmentation pipeline that was developed in-house. Volumes were correlated with clinical growth parameters at both the birth and time of MRI. **Results:** The CHD cohort showed significantly lower cerebellar volumes when compared with both the control (*p* < 0.015) and preterm (*p* < 0.004) groups. Change in weight from birth to time of MRI showed a moderately strong correlation with cerebellar volume at time of MRI (r = 0.437, *p* < 0.002) in the preterms, but not in the CHD neonates (r = 0.205, *p* < 0.116). Changes in birth length and head circumference showed no significant correlation with cerebellar volume at time of MRI in either cohort. **Conclusions:** Cerebellar development in premature-born infants is associated with change in birth weight in the early post-natal period. This association is not observed in term-born neonates with CHD, suggesting differential mechanisms of aberrant cerebellar development in these perinatal at-risk populations.

## 1. Introduction

Infants diagnosed with congenital heart disease (CHD) are at a higher risk for neurodevelopment impairment, with particular predilection for executive function and motor deficits [1,2,3,4,5]. While post-natal surgical intervention has been shown to increase this risk, it does not account for all of the observed deficits, suggesting a more complex perinatal mechanism of injury [6,7]. Of additional concern, term-born neonates with CHD are more likely to present with mild ischemic injuries, primarily periventricular leukomalacia (PVL) [8], and cerebellar cognitive affect syndrome [9], normally associated with preterm-born populations. These deficits significantly contribute to the longitudinal disease burden of CHD throughout the lifespan, with more individuals with CHD reaching adulthood than ever before [10,11,12]. 

Poor intrauterine somatic growth has been widely studied as a predictor of neurocognitive outcomes in preterm populations [13,14,15]. Asymmetrical intrauterine growth restriction where there was poor somatic growth, but brain sparing, was associated with better outcomes in preterm-born infants [16]. Importantly, preterm infants with favorable outcomes are more likely to demonstrate “catch up” growth with respect to both brain imaging metrics and neurodevelopment outcomes [15]. However, when studied in a single ventricle population, asymmetrical growth was not associated with better neurocognitive outcomes [17]. As such, understanding the interplay between somatic growth and brain development in CHD is imperative for prognosticating and for the longitudinal palliative care of individuals with CHD. Recent work in CHD has identified somatic growth restrictions as being potentially associated with post-operative mortality and long-term morbidity measured in early-term infants, even in slightly below average values for birth weight Z-scores [18].

In this work, we aim to explore the specific relationship between cerebellar development and somatic growth, as the cerebellum is a key structure that undergoes rapid development in the late gestation period, placing it at higher risk for injury [19]. Aberrant cerebellar development has been postulated in both preterm-born infants and infants with CHD [20]. Previous studies in infants with CHD have shown globally decreased brain volumes [21,22]; however, to the best of our knowledge, no study has specifically looked at cerebellar volume in relationship to somatic and head growth in this population. Our objective is to investigate the effect of CHD on cerebellar development, as well as its dependence on overall postnatal head and somatic growth. We compared term-born neonates with a history of CHD with a cohort of preterm-born neonates—two cohorts at risk for neurodevelopment impairment—to determine if they are similarly affected in this critical early developmental period. In Section 2, we describe in detail the subjects in each cohort (Section 2.1), the imaging acquisition parameters (Section 2.2), image processing methodology (Section 2.3), and statistical analysis (Section 2.4). In Section 3, we discuss the results and elaborate on the comparisons in both cerebellar volume across cohorts, and on the correlation between somatic growth and cerebellar volume at the time of MRI. In Section 4, we discuss the interpretation of our results and the potential mechanisms differently impacting otherwise healthy preterm-born infants and term-born infants with CHD in depth. Finally, we provide a brief conclusion and take-aways in Section 5.

## 2. Methods

### 2.1. Subjects

This work is an exploratory analysis of previously prospectively recruited neonates at our institution. We recruited 62 term-born CHD neonates, 54 term-born healthy neonates, and 51 preterm-born neonates as part of IRB-approved prospective studies. This dataset has been previously described in our prior work [23,24,25]. All imaging was reviewed by a clinical radiologist blinded to the diagnosis, and neonates with acquired brain injury (punctate lesions, infarcts, or hemorrhages) and syndromal issues were excluded from the analysis. The CHD inclusion criteria were only infants with a heart anomaly treated surgically in infancy, and included atrial septal defects (ASD), ventricular septal defect (VSD), hypoplastic left heart syndrome (HLHS), Ebstein’s anomaly, coarctation of the aorta (CoA), truncus arteriosus, transposition of the great arteries (TGA), and double-outlet right ventricle (DORV). Approximately 50% of the CHD cases were detected in utero and 50% were ex utero.

The mean gestation age for the control cohort was 41.2 (+/− 3.8), CHD cohort was 38.0 (+/− 2.9), and preterm cohort was 30.7 (+/− 3.8). Post-menstrual age (PMA) at time of imaging was 43.5 (+/− 5.5) for the control group, 42.4 (+/− 6.9) for the CHD group, and 40.5 (+/− 7.8) for the preterm group. Preoperative research brain imaging was conducted when the cardiothoracic intensive care unit (CTICU)/cardiology team determined the patient was stable for transport to the MRI scanner. A postoperative research scan was performed when the patient was younger than 3 months of postnatal age, either as an inpatient or outpatient. Most of our scans were research indicated and, as such, no additional sedation/anesthesia was given for the purpose of the scan. Most of the preoperative scans were performed on nonintubated, nonsedated patients; however, if a patient was intubated and sedated for clinical reasons at the time of the scan, their clinically indicated sedation continued under care of the primary CTICU team. Most of the postoperative scans were performed after the infant had stepped down from the CTICU and were done as “feed and bundle” scans without sedation.

The CHD and preterm cohorts had head and somatic growth measures taken at both birth and time of MRI, which included weight (WT), length/height (LT), and head circumference (HC). From these measures, we also calculated the relative head growth (change in HC/change in WT), adjusted for the time interval between birth and MRI. Head and somatic growth measures were not available for the term neonates; therefore, they were excluded from the growth rate analysis.

In order to account for the expected rapid growth in the preterm population (between birth and term equivalent age), we converted the raw growth parameters into Z-Scores for age, following established WHO guidelines [26] and normalized for time interval (zMRI-zBirth/Δtime). We calculated z-scores for weight-for-age (WAZ), length-for-age (LAZ), weight-for-length (WLZ), and head-circumference-for-age (HCAZ). From these, we also generated a measure of asymmetric growth, calculated as WAZ—HCAZ at both birth and MRI time points.

### 2.2. MRI Acquisition

Infants were scanned on a 3T Siemens Skyra MRI (Siemens, Erlangen, Germany) without sedation. Effort was made to scan at close to term equivalent age, or when infants were deemed clinically stable. Where available, we chose the earliest pre-operative scan for CHD infants. When no pre-operative imaging was possible, or the pre-operative scan failed QC, we used the earliest post-operative time point. The imaging parameters were as follows: (1) volumetric T1 magnetization-prepared rapid gradient-echo at echo time (TE)/repetition time (TR): 418/3100 ms, 1.0 × 1.0 × 1.0 mm^3^, and matrix size 320 × 320; (2) volumetric T2 sampling perfection with application optimized contrasts using different flip angle evolution sequence at TE/TR: 2.56/2400 ms, 1.0 × 1.0 × 1.0 mm^3^, and matrix size 256 × 196; (3) axial T2 weighted fast spin echo (FSE) at TE/TR: 3/2400 ms, slice thickness 2.0 mm with 0 skip, and in-plane matrix resolution 200 × 256. This dataset has been described in further detail in a previous publication [24]. Only infants who completed the volumetric imaging portion of the protocol were included in this study.

### 2.3. Structural Segmentation

We developed a semi-automated neonatal brain segmentation pipeline (Appendix A). This pipeline has been previously used to pre-process neonatal structural MRI for the classification of dysplastic cerebelli using a convolutional neural network [27]. The pipeline uses FSL’s brain extraction tool (BET) and FAST segmentation [28] to extract the brain and perform bias correction on the initial T2 image. The pipeline is then split into two complimenting branches. Branch 1 transforms the subject’s brain into an existing PMA matched probabilistic atlas [29] using the Advanced Normalization Tools (ANTS) algorithm [30]. This atlas is a set of neonate specific templates at various gestation ages containing six tissue probability maps: CSF, cortex, white matter, deep grey matter, brainstem, and cerebellum. The resulting transformation is then inversed, and the tissue probability maps are brought back into the subject’s native space. The final tissue volumes are calculated by binarizing each tissue map at a user defined probability threshold. Branch 2 of the pipeline uses ANTS to transform pre-parcellated, gestation age matched ALBERT subjects [31] into native subject space. We then performed a voxelwise winner-takes-all calculation to derive the binarized segmentation comprised of 50 non-overlapping regions. A manual correction step is required to ensure proper image registration and label generation. As each branch of the pipeline outputs a cerebellar volume, it presents a convenient tool for quality control. For this analysis, we used the final, manually corrected cerebellar volume from branch 2. The code for the pipeline is freely available at https://github.com/PIRCImagingTools/NeBSS (accessed on 5 July 2022).

### 2.4. Statistical Analysis

Statistical analysis was done in R (version 4.0.3, R Core Team) [32]. We compared the raw and normalized z-score somatic and head growth parameters between the CHD and preterm cohorts both at birth and time of MRI. We then looked at the change in these parameters between both time points, adjusting for the variable time interval between subjects. Finally, the PMA adjusted cerebellar volumes were correlated with clinical growth parameters at birth and time of MRI between the CHD and preterm cohorts. Comparison between cohorts was performed with ANOVA and Tukey’s HSD was used to identify between group differences when comparing the control, preterm, and CHD cohorts. The correlation between growth parameters and cerebellar volumes was calculated using Pearson’s r coefficient.

## 3. Results

There was no significant difference in PMA between the three cohorts at the time of imaging. Of the 62 CHD subjects, 11 did not have imaging pre-surgical intervention, and post-surgical intervention imaging was used. There was no statistically significant difference between pre- and post-op in the cerebellar volumes or growth parameters.

We looked at the standardized growth parameters for each cohort at both birth and time of MRI, comparing the CHD and preterm cohorts (Figure 1). Preterm-born neonates fell several standard deviations below the median values, while CHD neonates were only marginally below median values. By the time of MRI, the preterm neonates approached median values, while the CHD neonates stagnated or fell further below median values. Table 1 shows the difference between normalized growth parameters from birth to time of MRI, between the preterm and term CHD groups. Preterm-born neonates showed a general positive increase in z-score, while CHD neonates showed a continued decrease in growth parameter for age.

Appendix A shows the preterm group’s PMA adjusted cerebellar volumes, mean 699.3 (+/−3512.3) mm^3^, compared with the term controls, mean 457.8 (+/−2064.2) mm^3^ and the CHD group, mean −683.8 (+/−2428.0) mm^3^. The preterm cerebellar volumes were significantly higher than the CHD group (*p* < 0.008), and the CHD group also demonstrated a reduced cerebellar volume compared with the controls. Change in weight from birth to time of MRI showed a moderately strong correlation with cerebellar volume at the time of MRI (r= 0.437, *p* < 0.002) in the preterms, but not the in the term-born CHD neonates (r = 0.205, *p* < 0.116), significant at *p* < 0.000 (Figure 2). However, when normalized to Z-scores, no significantly strong correlation was observed. Changes in both birth length and occipital-frontal circumference (from birth to time of MRI) showed no significant correlation with cerebellar volume at the time of MRI in either cohort.

## 4. Discussion

In this study, we show that within the first weeks post-birth, term-born CHD neonates demonstrate reduced PMA adjusted cerebellar volumes compared with healthy controls and term-equivalent aged preterm-born neonates. It is well known that preterm-born infants exhibit accelerated somatic growth (catch up period) after birth [33]. Here, we observed that while preterm individuals reached normative volumes within weeks (adjusted for PMA), CHD neonates remained below normative values. This is consistent with a study that observed no catch-up growth seen after cardiac surgery, and found a high prevalence of persistent microcephaly associated with long-term neurologic and psychomotor impairment in children with CHD [17]. Importantly, we observed no relationship between cerebellar volume and postnatal growth in term CHD, in contrast with an observed relationship between birth weight and preterm cerebellar development.

Head circumference and somatic growth are useful measures that have been thoroughly validated in preterm populations [14,34], and they moderately correlate with short-term outcomes in preterm development [35]. Efforts have been made to determine whether they are predictive of longitudinal neurocognitive outcomes, but are confounded by compensatory processes and environmental factors [36]. However, less is known about their prognostic utility in CHD populations. A recent study showed that children with CHD exhibit only partial catch-up in somatic growth measured at 10 years of age, indicative of a persistent developmental delay through childhood [37]. Investigations into the mechanistic underpinnings of aberrant brain development in CHD indicates that there are more complex underlying mechanisms driving poor outcomes in single ventricle infants [38]. Correlating growth measures with structural brain volumes may provide further insight into these mechanisms. Work in CHD fetal imaging shows smaller brain structure volumes detected in utero, with associated impaired neuroaxonal development and abnormal metabolism [39]. Additionally, domain specific functional deficits have been correlated with reduced structural volumes in neonates with CHD, indicating that volumetric MRI can be a useful screening tool in CHD [40,41]. CHD infants are more likely to exhibit impaired growth, independent of focal white matter injury, preferentially in frontal lobes and brainstem measured prior to surgical intervention [42], as well as altered structural connectivity [43]. In contrast, in preterm infants, a longitudinal decrease in brain volume, increase in pituitary height, and thinning of corpus callosum have been observed, but only in more mature preterm infants with severe growth restriction well past birth [44]. Thus, the pattern of growth impairment in CHD across the lifespan is heterogeneous with respect to HC, weight, and height, which are inconsistently affected. These dissimilarities may be related to differences in cardiac anatomy, demographics, medical factors, and socio-economic status.

While studies in CHD patients do show catch-up growth in height and, to a lesser extent, in weight and head circumference by 10 years of age, somatic growth can still remain impaired with respect to head circumference and overall weight z-scores below the norm for up to 10 years [37]. The etiology of this impaired growth is likely related to multiple factors, such as reduced preoperative blood oxygenation and perfusion, reduced substrate delivery, age at surgery, neurological abnormality including seizure activity and overall duration of hospitalization, and increased medical complexity including increase number of cardiac catheterizations. We recently described important associations between subcortical (thalamic and cerebellum) morphological reductions and altered cerebral white matter metabolism in both preterm and term CHD infants [45,46]. We anticipated that these subcortical morphological reductions, which were also associated with other global measures of cerebral structures (as measured with brain metrics), would be associated with global alterations in brain maturation metabolites (NAA, choline, and myo-inositol), which normally change dramatically with age during the time period considered in our study [47]. We previously found that reduced cerebellar volume, most pronounced in a preterm CHD group, was associated with reduced glutamine in parietal grey matter in both CHD groups [20]. Single ventricle, obstructed aortic arch, and cyanotic types of CHD lesions were predictive of the relationship between reduced subcortical morphometry and reduced GLX (mostly glutamine) in both CHD cohorts (frontal white matter/parietal grey matter). Subcortical morphological associations with brain metabolism were distinct within each of the three groups, suggesting that these relationships in the CHD groups were not directly related to prematurity or white matter injury alone.

Here, we chose to focus specifically on cerebellar volumes and its correlation with growth parameters. Cerebellar abnormalities have strong associations with neurocognitive outcomes in both preterm and CHD populations [48]. A potential mechanism of injury has been investigated as the sonic hedgehog (SHH) mediated mechanism of cerebellar hypoplasia in prematurity [49]. Incidentally, SHH may also play a role in cerebellar development in CHD [23,50]. Our recent work in brain development shows infants with CHD are more likely to present with cerebellar dysplasia [24], which may correlated with ciliary dysfunction [23], which is associated with executive function disorders [51]. However, our findings suggest possibly different dynamics of aberrant cerebellar development between these populations. Complex hemodynamic and metabolic factors may be important, and further study is needed [52]. The cerebellum has traditionally been associated with motor control and balance, including the ability to master complex motor movements. However, there is more recently evidence that the cerebellum is important for important cognitive functions, including emotional regulation [53,54]. The cerebellum has multiple features that are relevant to the neurodevelopmental vulnerabilities in congenital heart disease, including (1) protracted developmental trajectories that extends during the last half of gestation and first couple of postnatal months; (2) sexual dimorphism; (3) key vulnerability to environmental exposures; and (4) frequent associations with autism and ADHD, which can all be seen in patients with congenital heart disease. Importantly, ex-premature infants with cerebellar injury are at higher risk for complex neurologic sequelae, which are characterized by cerebral cortical dysfunction, very district from motor dysfunction [55]. In older subjects with cerebellar injury and cerebellar malformations, this presentation of nonmotor cognitive dysfunctional sequelae is named cerebellar cognitive affective syndrome [9]. While this type of developmental cerebellar cognitive affective syndrome has been clearly shown in in survivors of prematurity-related cerebellar injury, this presentation has not been clearly demonstrated in pediatric patients with congenital heart disease. It is likely that the underpinnings of a potential developmental cerebellar cognitive affective syndrome in congenital heart disease patients may be related to stressors and exposures (including reduced substrate delivery and hypoxia) during critical periods of cerebral cortical development in the third trimester and infancy. The cerebellum is an important brain area that is increased in size in humans compared with other mammals and it evolved late in evolution. This is a robust increase in the size of the cerebellar hemispheres when compared with the midline vermis, which is known to be the predominate component of the cerebellar cortex in many mammals, and is also known be phylogenetically older than any other component of the cerebellar cortex compared with other mammals. The cerebellar cortex is also known to have cytoarchitectural uniformity of its cortex when compared with cerebral cortical layers in different supratentorial lobes of the brain. Cerebro−cerebellar loops are an important component of functional specificity of different brain regions, which also show a high degree of conservation across species [56]. The biggest functional subdivision of the cerebellum is the medio−lateral division in the cerebellum, which is located between the vermis and the hemispheres. Another important loop is the lateral portions of the cerebellar hemispheres, which are inter-connected to the prefrontal cortex which is underlying the important role of the cerebellum in cognitive processes. The vulnerability of these cerebellar−cortical motor and cognitive loops in congenital heart disease patient is not well delineated, particularly in relation to somatic growth parameters, as measured in our current study.

This work has some limitations. Due to the precarious condition of infants born with CHD, there is some variance in the time interval between birth and MR imaging. We attempted to mitigate this by calculating a growth rate measure, at a trade-off of the direct interpretability of these values. Furthermore, a convenience sample cannot be avoided, as patients not stable enough to image were excluded from the analysis. There may be bias due to the less-stable cases not getting pre-surgical MRI, but there are also uncorrelated reasons, including scheduling difficulties, motion artifacts/failed scans, or consent to study done only after surgery. We used the pre-operative time point for CHD infants whenever possible, and limited the time interval post-operative to mitigate as much of potential surgically related changes as we could. Unfortunately, it was not possible to image every neonate prior to surgery, but we feel that including post-operative infants within a short time frame would still primarily reflect pre-operative findings. We believe this is not directly detrimental to our overall hypothesis, as characterizing aberrant development in relatively healthy infants with CHD points to these deficits being implicit in CHD, independent of clinical condition at birth. Additionally, as this was a secondary analysis of a prospective study, we did not power recruitment to specifically look at growth changes from birth to MRI, and did not control for nutrient accretion. Similarly, we did not acquire growth data on the control population, and the reported gestational age of the controls (as taken from the available clinical record) may be biased—where term-born infants may be defaulted to 40 weeks gestation. However, the primary purpose of this work was to contrast preterm and CHD infants, and comparisons to the control population were only used as a comparative baseline. We believe using normalized values and z-score conversions helps attenuate this shortcoming, but further investigation is necessary to confirm our findings.

Future work aims to more granularly differentiate between serial pre-operative and post-operative MRI scans in infants with CHD. Additionally, we will incorporate socio-economic status, genetic data, and clinical factors associated with perinatal injury into our statistical models to further investigate the short- and long-term neurodevelopmental outcome correlates of our cerebellar morphological findings.

## 5. Conclusions

Cerebellar development in premature-born infants is associated with change in birth weight in the early post-natal period. This association is not observed in term-born neonates with CHD, suggesting differential mechanisms of aberrant cerebellar development in these perinatal at-risk populations. Taken together, our findings are consistent with a body of work showing that somatic and head growth are important biomarkers that may reflect deeper mechanistic brain development in infants with CHD.

## Figures and Tables

**Figure 1 diagnostics-12-01644-f001:**
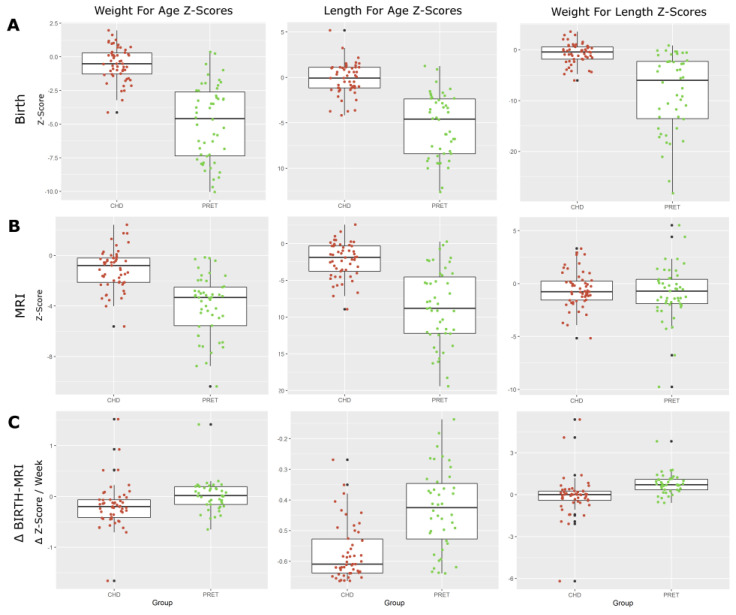
Growth parameter z-scores at (**A**) birth, (**B**) time of MRI, and (**C**) change between birth and time of MRI. WAZ, LAZ, and WLZ at birth shows the expected lower z-scores for preterm-born neonates. CHD neonates at birth show slightly below average z-scores. At the time of MRI, we start to see preterm-born neonates approach normative values. The changes in WAZ, LAZ, and WLZ from birth to MRI show a consistent decrease in CHD neonates, while preterm-born neonates show an increase in weight for age and weight for length, and a decrease in length for age.

**Figure 2 diagnostics-12-01644-f002:**
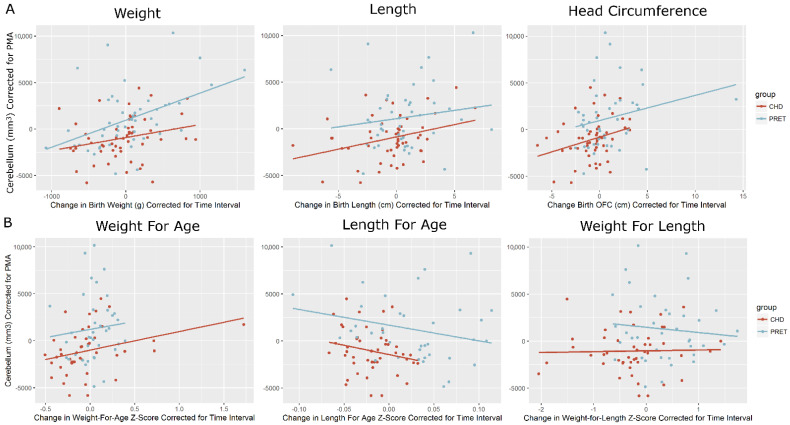
Cerebellar volumes (corrected for PMA) correlated with growth parameters at time of MRI. (**A**) Raw measurements and (**B**) z-Scores Growth parameters were converted to normative Z-scores following WHO guidelines, and normalized for time interval (zMRI-zBirth/Δtime). Volumes were corrected for post-menstrual age (PMA). Change in weight from birth to time of MRI showed a moderately strong correlation with cerebellar volume at time of MRI (r = 0.437, *p* < 0.002) in the preterms, but not the in the term-born CHD neonates (r = 0.205, *p* < 0.116), significant at *p* < 0.000. Changes in both the birth length and occipital-frontal circumference (from birth to time of MRI) showed a moderate (but not significant) correlation with cerebellar volume at the time of MRI in both cohorts. When normalized to Z-scores, no correlation is observed.

**Table 1 diagnostics-12-01644-t001:** Change in standardized growth parameter Z-Scores from birth to time of MRI. Each growth parameter converted to a standardized z-score and normalized for time interval. Preterm-born neonates show a general positive increase in z-score, while CHD neonates show a continued decrease in growth parameter for age.

	CHD	Preterm	
Change in Metric (Birth—MRI)	Mean (SD)	Mean (SD)	(*p*<)
Weight for Age Z-Score (WAZ)	−0.18 (0.44)	0.02 (0.31)	0.012
Length for Age Z-Score (LAZ)	−0.57 (0.09)	−0.43 (0.13)	0.000
Weight for Length Z-Score (WLZ)	−0.08 (1.48)	0.76 (0.75)	0.001
Head Circumference for Age Z-Score (HCAZ)	−0.14 (0.87)	0.03 (0.52)	0.263
Asymmetry (WAZ—HCAZ)	−0.03 (0.80)	−0.01 (0.45)	0.898

## Data Availability

Limited data available on request (pending Data Usage Agreement) due to restrictions on subject privacy and data anonymization.

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
