# Peer review of "Reduced Cerebellar Volume in Term Infants with Complex Congenital Heart Disease: Correlation with Postnatal Growth Measurements"

_diagnostics, 2022, doi:10.3390/diagnostics12071644_

Round 1

Reviewer 1 Report

This paper studies Reduced Cerebellar Volume in Term Infants with Complex Congenital Heart Disease: Correlation with Postnatal Growth Measurements. This is an interesting paper that could be a potentially publishable subject. There are a few weaknesses that should be addressed in this paper. Therefore, I suggest the authors resubmit it after a major revision. My suggestions are as follows:

General comments:

1. The structure of your paper is so weird. Please consider the main structure of the paper at the end of the introduction.

2. Discuss more the limitations of the study and future research suggestions if there are any.

3. The paper should be revised to include more recent references in 2021-2022.

4. Your section and subsections should be ordered in numbers based on the MDPI structure.

5. Please consider a conclusion. You only provided a discussion.

6. The quality of English needs to be improved across the paper. Also, the scientific terms pertinent to your topic should be improved.

 Main comments:

1. In lines 192-193 you have mentioned "Growth parameters were converted to normative Z-Scores following WHO guidelines, and normalized for time interval"  Please explain more about conversion and normalization? 

2. In lines 195-198 you mentioned  "Changes in both 195 length and occipital-frontal circumference (from birth to time of MRI) showed no significant correlation with cerebellar volume at time of MRI in either cohorts. However, when normalized to Z-scores, no significantly strong correlation is observed"

Why when normalized to Z-scores, no significantly strong correlation is observed? Is there any other option please clarify more?

3. Please explain more about “MRI Acquisition” in line 109.

4. In lines 125-126 you have mentioned "We developed a semi-automated neonatal brain segmentation pipeline (Supple- 125 mental Figure 1)" Please explain more about the neonatal brain segmentation pipeline.

5. In lines 149-150 "Comparison between cohorts was performed with ANOVA and Tukey’s HSD (where applicable), and correlations used Pearson’s r" Why did you just provide the suggested approach? please explain more about your statistical part

Author Response

Reviewer 1:

Comments and Suggestions for Authors:

 This paper studies Reduced Cerebellar Volume in Term Infants with Complex Congenital Heart Disease: Correlation with Postnatal Growth Measurements. This is an interesting paper that could be a potentially publishable subject. There are a few weaknesses that should be addressed in this paper. Therefore, I suggest the authors resubmit it after a major revision.

My suggestions are as follows: General comments:

  1. The structure of your paper is so weird. Please consider the main structure of the paper at the end of the introduction.

Thank you for the suggestion. We have revised our structure to fit the MDPI standard.

  1. Discuss more the limitations of the study and future research suggestions if there are any.

Thank you. We have added a more thorough discussion of the limitations of the work and further discussion on future directions (p. 11).

  1. The paper should be revised to include more recent references in 2021-2022.

Thank you. We have added more recent references.

  1. Your section and subsections should be ordered in numbers based on the MDPI structure.

Thank you for the suggestion. We have revised our structure to fit the MDPI standard.

  1. Please consider a conclusion. You only provided a discussion.

Thank you for the suggestion. We have added a conclusion section.

  1. The quality of English needs to be improved across the paper. Also, the scientific terms pertinent to your topic should be improved.

Thank you. We have done a thorough re-read and cleaned up the language.

Main comments:

  1. In lines 192-193 you have mentioned "Growth parameters were converted to normative Z-Scores following WHO guidelines, and normalized for time interval" Please explain more about conversion and normalization?

Thank you for the suggestion. We have provided further details in the methods (p. 4) as well as figure 2 legend.

  1. In lines 195-198 you mentioned "Changes in both 195 length and occipital-frontal circumference (from birth to time of MRI) showed no significant correlation with cerebellar volume at time of MRI in either cohorts. However, when normalized to Z-scores, no significantly strong correlation is observed" Why when normalized to Z-scores, no significantly strong correlation is observed? Is there any other option please clarify more?

Thank you for catching this. It is indeed awkward phrasing and we have clarified: we observed a moderate correlation (but not significant) when comparing changes in OFC, and this correlation is not observed when normalized to Z-scores.

  1. Please explain more about “MRI Acquisition” in line 109.

Thank you for the suggestion. We have previously published a more detailed description of this dataset, and have included the citation (p. 5)

In lines 125-126 you have mentioned "We developed a semiautomated neonatal brain segmentation pipeline (Supple- 125 mental Figure 1)" Please explain more about the neonatal brain segmentation pipeline.

Thank you for the suggestion. We have previously used this pipeline to pre-process neonatal structural MRI, and described the pipeline in more detail in a previous publication. We have included a citation to the prior work (P. 5).

  1. In lines 149-150 "Comparison between cohorts was performed with ANOVA and Tukey’s HSD (where applicable), and correlations used Pearson’s r" Why did you just provide the suggested approach? please explain more about your statistical part

We have expanded this to provide clarification for the choice of statistical tests.

Reviewer 2 Report

The study concept is valuable, as this further adds to the field on available knowledge on the cerebellar growth in different neonatal populations.

The study design reads somewhat unanticipated. I therefore would like to known whether this study was predefined by a protocol, or is this rather ‘pragmatic’ in its design (explorative). The methods section suggest prospective data collection, but eg a power calculation is not mentioned.

How do you cover the anticipated postnatal weight loss, before accretion occurs

There is already quite some literature that the brain in CHD cases is already ‘different’ before surgery, so how has this been integrated in this study and its discussion ? I understood this pre-surgery literature as if the CHD brain (especially in left output track issues, or TGV) reflects an proportionally ‘immature’ brain.

Was the radiologist blinded for group allocation, and how do you discriminate ‘acquired brain injury’ as the delayed growth is likely also acquired ?

CDH MRI imaging was acquired either before, or after surgery, but is this in indeed not another bias, as the postsurgical cases will likely be the less stable cases ?

As scans were suggested to be collected as close a possible to term equivalent age, this is a discordant time compared to the CHD cases (unless this refers to the pre surgery scans ?) If so, this should be made clearly, but when collected in early neonatal life in CHD cases, the early postnatal weight loss becomes an even more relevant issue. This is also presented as a mitigation strategy in the discussion, but if you take an MRI shortly after birth, the weight will be likely lower compared to the birth weight. Perhaps a straight forward PMA matched analysis with Z scores at PMA are therefore more appropriate instead of an extrapolation of trends over postnatal life, with significant differences in duration of PNA, as this in itself could already affect the robustness of the Z trends ? How old were the CHD cases (pna) versus the former preterm cases ,

Related to the ‘timing’, I suggest to further consider the first sentence of the discussion as I assume that the ‘first weeks post birth’ do not apply to the preterm neonates ?

I miss information on the number of cases considered, the number of cases recruited, and the number of cases retained in the study (line 122, likely more likely to occur in non-sedated cases ?). Furthermore, are the any CHD cases with ‘syndromal issues’, and what type of CDH were included ? were all cases detected before delivery, or some after ‘postnatal’ screening versus ‘post shock’ setting ?

Are you sure on the mean gestation age in controls (subject section), as 41.3 weeks is quite high, and the SD suggested is likely not a normal distribution ?) (or are these PMA at imaging ?) I suggest to report the ranges as this will likely better inform the readership on the characteristics of your dataset.

What is the add on value of the healthy controls, as these data are not reported in the core paper, but are provided in the supplement ? is this because the reference values of the healthy cases is already reported, or will be reported elsewhere. I’m ok with both options, but eg the current figure 1 is also restricted on preterm and CHD cases.

Minor

I think that the ‘editing’ of the paper needs some additional adaptations to be in line with the journal’s policy.

Author Response

Reviewer 2:

Comments and Suggestions for Authors

  1. The study concept is valuable, as this further adds to the field on available knowledge on the cerebellar growth in different neonatal populations. The study design reads somewhat unanticipated. I therefore would like to known whether this study was predefined by a protocol, or is this rather ‘pragmatic’ in its design (explorative).

This is a good suggestion, thank you. We have clarified in our study design that this was an explorative analysis of data acquired through other prospective studies specific to preterm development and/or congenital heart disease, and have cited the relevant prior work (p.3).

  1. The methods section suggest prospective data collection, but eg a power calculation is not mentioned. How do you cover the anticipated postnatal weight loss, before accretion occurs

Thank you again. Similar to the above suggestion, we clarify that the primary purpose of the data collection was not to look at growth parameters. We have added this as a limitation of the study.

  1. There is already quite some literature that the brain in CHD cases is already ‘different’ before surgery, so how has this been integrated in this study and its discussion ? I understood this presurgery literature as if the CHD brain (especially in left output track issues, or TGV) reflects an proportionally ‘immature’ brain.

This is a good point, and what we are attempting to investigate. We used the pre-operative time point for CHD infants wherever we can, and limited the time interval post-operative to mitigate as much of potential surgically related changes as we could. Unfortunately, it was not possible to image every neonate prior to surgery, but we felt that including post-operative infants within a short time frame would still primarily reflect pre-operative findings within the literature. We have further clarified this limitation in the discussion.

  1. Was the radiologist blinded for group allocation, and how do you discriminate ‘acquired brain injury’ as the delayed growth is likely also acquired ?

Thank for this suggestion. We have included more information on both the CHD types, inclusion/exclusion, and radiologist review in our Methods section (p. 3). The radiologists were indeed blinded, and specifically looked for punctate lesions, infarcts, or hemorrhages.

  1. CDH MRI imaging was acquired either before, or after surgery, but is this in indeed not another bias, as the postsurgical cases will likely be the less stable cases ? As scans were suggested to be collected as close a possible to term equivalent age, this is a discordant time compared to the CHD cases (unless this refers to the pre surgery scans ?) If so, this should be made clearly, but when collected in early neonatal life in CHD cases, the early postnatal weight loss becomes an even more relevant issue. This is also presented as a mitigation strategy in the discussion, but if you take an MRI shortly after birth, the weight will be likely lower compared to the birth weight. Perhaps a straight forward PMA matched analysis with Z scores at PMA are therefore more appropriate instead of an extrapolation of trends over postnatal life, with significant differences in duration of PNA, as this in itself could already affect the robustness of the Z trends ?
    This is indeed a limitation of this study. There may be bias due to the less-stable cases not getting pre-surgical MRI, it is not always the case that it is due to clinical condition – it may also be due to scheduling difficulties, motion artifacts/failed scans, or consent to study done only after surgery. We also limit the time interval allowed after surgery to mitigate time-related confounders. Ultimately, we chose to use time normalized Z-scores to hopefully mitigate this issue as much as possible. Using a static time-point value (z-score adjusted or not) would not answer the specific growth-restriction question we are trying to answer in this study, and the variance in PMA at time of scan between cohorts would be an additional confounder. We have expanded the limitations section to be more clear regarding this limitation. (p.11)
  2. How old were the CHD cases (pna) versus the former preterm cases , Related to the ‘timing’, I suggest to further consider the first sentence of the discussion as I assume that the ‘first weeks post birth’ do not apply to the preterm neonates ?
    Thank you for this. The sentence in question is indeed confusing, as it is referring to CHD cases specifically (not preterm infants). We have clarified this in the discussion (p. 7)

  3. I miss information on the number of cases considered, the number of cases recruited, and the number of cases retained in the study (line 122, likely more likely to occur in non-sedated cases ?).
    This is a good observation, and we have clarified the recruitment strategy. As this is an exploratory analysis of data collected from multiple prospectively recruited studies, we did not include the full number of cases considered/recruited as it does not directly reflect the selection method for this analysis. We have included references to prior work investigating each study cohort independently (p.3).
  4. Furthermore, are the any CHD cases with ‘syndromal issues’, and what type of CHD were included ?

Thank you for the suggestion. Syndromal issues were excluded and we have clarified in our methods section.

  1. were all cases detected before delivery, or some after ‘postnatal’ screening versus ‘post shock’ setting?
    From the clinical record, about 50% were detected in utero, 50% ex utero. We have added this to the cohort description (p. 3)
  2. Are you sure on the mean gestation age in controls (subject section), as 41.3 weeks is quite high, and the SD suggested is likely not a normal distribution ?) (or are these PMA at imaging ?) I suggest to report the ranges as this will likely better inform the readership on the characteristics of your dataset.

This is a good observation. We have checked the clinical records and confirmed this to be accurate (per the clinical chart). There may be a charting bias (where term infants are charted at 40 weeks by default). As this method would not provide an accurate range, we have kept the SD in the manuscript and added this to the limitations section. As the term infants are primarily serving as a baseline comparison for overall cerebellar volume, we feel this does not affect the main conclusions of the manuscripts.

  1. What is the add on value of the healthy controls, as these data are not reported in the core paper, but are provided in the supplement ? is this because the reference values of the healthy cases is already reported, or will be reported elsewhere. I’m ok with both options, but eg the current figure 1 is also restricted on preterm and CHD cases.

We included a comparison to controls only for the cerebellar volumes, as the control infants did not get growth measures taken as part of the research protocol.

  1. Minor I think that the ‘editing’ of the paper needs some additional adaptations to be in line with the journal’s policy.

Thank you. We have done a thorough revision, cleaned up the language, and formatted to be in line with the MDPI guidelines.

Round 2

Reviewer 1 Report

The authors just answered all suggested comments exactly except the first one which was:

1. Please consider the main structure of the paper at the end of the introduction. For example:

At the of the introduction, you should mention:

In part 2 the methods are proposed including subjects, MRI Acquisition, structural segmentation, and statistical analysis.....in part 3 results are discussed which elaborate.....

By mentioning this structure at the end of the introduction, you will provide an outline for readers. 

Author Response

1. Thank you for clarifying. We have included at the end of the introduction a roadmap to the remaining structure of the paper.

Reviewer 2 Report

no additional comments, 

Author Response

Thank you. We appreciate your thorough review.